# Working Memory and Language Relate to Report of Socio-Emotional Functioning in Children with Hearing Loss

**DOI:** 10.3390/jcm13061637

**Published:** 2024-03-13

**Authors:** Dorothy A. White, Elizabeth Adams Costa, Nancy Mellon, Meredith Ouellette, Sharlene Wilson Ottley

**Affiliations:** The River School, 4880 MacArthur Boulevard, Washington, DC 20007, USA; eadams@riverschool.net (E.A.C.); nmellon@riverschool.net (N.M.);

**Keywords:** hearing loss, childhood, working memory, language, socio-emotional

## Abstract

**Background:** Children with hearing loss have been found to have significantly more behavioral and emotional challenges than their typically hearing peers, though these outcomes are variable at the individual level. Working memory deficits have been found to relate to executive functioning and overall emotion regulation, leading to behavior challenges. Language development is essential for development of social relationships and communicating one’s needs and this may lead to distress when children cannot communicate effectively. Based on prior findings in children with hearing loss and their typically hearing peers, working memory and language skills were hypothesized to be related to parent and teacher report of socio-emotional functioning. **Methods:** Participants were 35 children with hearing loss (66% female, M = 5.17 years old, SD = ±1.97) whose language, working memory, and socio-emotional functioning were evaluated during the course of treatment and educational planning. **Results:** Bivariate analyses indicated that working memory was related to a number of socio-emotional domains (e.g., functional communication, atypicality, withdrawal), as were language scores (e.g., social skills, inattention). The direction of these associations was such that stronger working memory and language skills were related to more regulated socio-emotional functioning. **Conclusions:** This study is limited in generalizability by size and the relative homogeneity of the sample. A call to action of the current study includes more education with regard to profiles and presentations of children with hearing loss, and an early focus on socio-emotional learning to foster the development of regulatory skills.

## 1. Introduction

Early auditory deprivation can have a multitude of effects on brain development and may impact cognitive capacities that extend beyond the auditory system [1]. Hearing loss can be considered a connectome disease, with individual differences in response to that auditory deprivation accounting for variability in outcomes [2]. The existing body of research indicates that areas of development are interconnected, such that deficits in one area of development have cascading effects on others. By extension, loss of auditory input would impact a variety of later outcomes, including cognitive, linguistic, and socio-emotional functioning. This is corroborated by findings indicating a significantly higher rate of socio-emotional and behavioral problems in children with hearing loss [3], and differences in neural dynamics and processing when compared to typically hearing peers [4].

To better understand the relationships between these areas of development in children with hearing loss, outcomes and cognitive profiles for these children should be examined. Published literature on outcomes for children with cochlear implants (CIs) and hearing aids has been highly variable, with outcomes for children with hearing loss being similar, poorer, or stronger than those of their typically hearing peers depending on the domain [1,3,5,6,7]. Studies investigating the cognitive profiles of children with hearing loss document this variability; some studies report lower scores on measures of verbal abilities [1,5,6,7] and working memory tasks [1,8,9] compared to typically hearing peers, while others report commensurate scores on similar measures [1,4]. Overall, prior research regarding outcomes and the impact of associated factors is inconsistent and merits further investigation. Due to the inconsistency of these findings in the literature, further investigation is warranted.

Existing literature has highlighted the relationship between working memory and language in children with hearing loss, supporting the need for further investigation of cognitive and language-based characteristics [4,8,9]. Relationships have been found between working memory and language development in children with CIs: working memory is acknowledged to be critical to the development of speech and language [8,9]. Children with hearing loss have demonstrated varying degrees of speech comprehension and understanding; associated factors include age of implantation, device wear time, nonverbal cognitive ability, parental education, and socio-economic status, among others [5,10]. Understanding if, and how, these impacts are similar or different across children with hearing loss can aid in the development of targeted and specific early intervention programs. The current study seeks to document specific cognitive, linguistic, and socio-emotional factors that have a significant impact on development in the population of children with hearing loss.

### 1.1. Factors Influencing Language Development

Hearing loss status is a significant factor in language development in childhood. Prior research has found that there is variability in how language skills develop in children with hearing loss [1,5,6,10]. Specifically, lower scores in expressive and receptive language use have all been noted when children with hearing loss are compared to their typically hearing peers [10]. Due to a lack of auditory input in infancy, children with hearing loss have less language feedback, which impacts their ability to develop appropriate language skills [2,5]. In the first year after cochlear implantation, wear time of listening devices significantly predicted receptive language outcomes, suggesting that the increased exposure to auditory stimuli positively supports language growth [5]. While a gap is observed in expressive and receptive language skills between children with hearing loss and their typically hearing peers due to this early auditory deprivation, early intervention services have been shown to close this gap [11].

Differences have been noted in outcomes for children based on mode of amplification, degree of hearing loss, and type of hearing loss. Children with mild hearing loss were found to have poorer phonological processing and literacy outcomes than their same-aged peers [7], and bilateral cochlear implant users were found to have stronger single-word vocabulary than children with unilateral implants and the same severity of loss [12]. Age of implantation has been found to be significantly related to expressive and receptive language outcomes, including more typical and faster early language development [5,13]. However, not all of the variance in expressive language outcomes can be accounted for by activation age, suggesting that further investigation into related factors is important. Other studies have indicated no relationship between age of implantation and later language outcomes, suggesting that more research into the predictive capacity of demographic variables is necessary [10].

Educational environment can also have a substantial impact on language acquisition in children with hearing loss. For example, the COVID-19 pandemic disproportionally impacted deaf and hard of hearing children because of auditory access, attention, reduced availability of individual teacher support, and ability to read provided captions on video calls, further restricting their access to academic content and literacy instruction [14]. Strong oral language ability has been linked to literacy development in studies of children with hearing loss, especially in conjunction with early intervention in language [6,7,13]. Overall, language is an essential area of skill development for children with hearing loss, and one that can shape many domains of functioning.

One hypothesis for why children with hearing loss struggle with language is that these children do not have the same benefit of “overhearing” and gaining incidental exposure to adult and peer language. As a result, their access to acquisition of knowledge and social skills can be limited [15]. Language growth can also be essential to development of social relationships: some children with hearing loss disengage in noisy or disruptive environments like classrooms and have difficulty participating if they cannot hear what is being said by peers [15]. Difficulties in speech and language development can have an impact on peer relationships, especially if communication with peers becomes challenging [15,16]. Additionally, some children with hearing loss act as shy and withdrawn, avoiding or minimizing peer relationships overall [16].

Self-esteem, mood, and social development can be vulnerabilities for children with hearing loss [15,16]. Better communication skills have been found to be associated with higher self-esteem and social competence, both of which have been linked to socio-emotional factors [15,16]. Self-consciousness or discomfort about wearing one’s amplification devices can impact device wear time, and thus incidental language exposure [16]. Ability to affiliate with a community and communicate have been described as relating to self-reported quality of life in children with hearing loss [16]. Noisy environments can cause dysregulation when children cannot rely on their devices for access, leading to frustration or sadness [16]. Overall, socio-emotional functioning is linked to language competence and skill development, and this relationship has been noted previously in children with hearing loss.

### 1.2. Working Memory and Hearing Loss

Working memory, and especially auditory working memory, has been found to be poorer in individuals with hearing loss when compared to their typically hearing peers [17,18]. In children with single-sided deafness, significant differences in working memory performance were observed between those with amplification (i.e., hearing aids or bone-anchored hearing aids) and those without [19]. Neuro-imaging studies in children with hearing aids have found that more typical neural functioning around encoding and maintenance across regions of the brain was related to duration of hearing aid use [20]. A relationship between working memory and hearing loss has been found, though more research is required to understand the mitigating factors of device use and type of hearing loss.

A current hypothesis regarding the relationship between working memory and hearing loss suggests that working memory is related to challenges around listening in noise, or the ability to distinguish target information in noisy environments [20,21]. Ability to listen effectively in noise has been found to be related to working memory precision, with more difficulty in listening in noise associated with more challenges in working memory [20,21]. Further research on the relationship between working memory and hearing loss is warranted, though it is possible that this relationship is driven by listening in noise.

### 1.3. Working Memory and Socio-Emotional Functioning

Working memory in early childhood has been previously identified as a predictor of socio-emotional functioning in later years [22]. It is likely that this relationship is driven by executive functioning, or the ability to self-regulate in the behavioral, cognitive, and emotional domains. Self-reported inattention, a domain associated with working memory and executive functioning, was associated with self-esteem in children with hearing loss, a variable that relates to many aspects of social ability, including number of friends, shyness, and likelihood of engaging in social activities [16]. Working memory training has broadly been found to support emotion regulation and mood in a variety of trials [22,23,24]. Stronger working memory is hypothesized to reduce cognitive load across a variety of diagnostic profiles (e.g., PTSD, depression, eating disorders), improving cognitive efficiency in areas such as emotion regulation [24]. Better understanding and confirmation of this relationship can support development of specific and targeted intervention. Challenges in working memory can lead to difficulty in retaining and applying academic and social information; without appropriate access to this information, children may become more dysregulated, leading to difficulty in school and in friendships.

### 1.4. Study Aims

This study was designed to better understand the cognitive, linguistic, and socio-emotional profiles of children with hearing loss. The primary aim of this work is to document the relationships between factors influencing development across domains in children with hearing loss. A secondary aim was to lay a foundation for future and prospective research by conducting initial analyses. The current study hypothesized that, among children with hearing loss, working memory capacity and language ability would predict socio-emotional functioning, with stronger working memory and language predicting better overall functioning. Specifically, it was expected that higher scores on the WISC-V or WPPSI-IV Working Memory Index scores would be related to and predict lower scores on the clinical scales of the BASC-3 Parent and Teacher Report. Additionally, it was hypothesized that higher language scores on expressive vocabulary, receptive vocabulary, overall language, and pragmatic language measures would also be related to and predict lower scores on the BASC-3.

## 2. Materials and Methods

### 2.1. Research Design

This study was conducted at a single time point and serves to describe the characteristics of the sample through correlation and linear regression. This descriptive design examines relationships between study variables across individuals. The current study is a retrospective design. All data used in the current study were not collected for research purposes. This study consisted of analysis of clinical data that was collected in its entirety prior to the onset of research. Thus, the current study was eligible to be performed without IRB approval or exemption as a post hoc analysis. This study complied with the 1975 Declaration of Helsinki and its subsequent amendments, or comparable ethical standards.

### 2.2. Participants

Participants were 35 children with hearing loss (66% female, M = 5.17 years old, SD = ±1.97) enrolled in a private, auditory–oral school that ran an inclusion program (i.e., children with typical hearing and children with hearing loss were taught together). All participants were in classrooms with a full-time speech language pathologist and master’s level educator within a co-teaching model. 94% of participants received additional auditory–verbal therapy or individual speech and language therapy at the time of evaluation.

Severity of hearing loss varied, with the majority of children either falling in the severe or profound ranges of hearing loss in at least one ear (71–90+ dB; see Table 1 for further audiological information). Audiological evaluations were conducted at least twice yearly for all participants, and more frequently if concerns about auditory access were noted. Daily “listening checks” are conducted, during which the child’s classroom speech-language pathologist checks the child’s hearing aids or cochlear implants in both unilateral conditions and the bilateral condition.

### 2.3. Procedure

Participants were administered annual cognitive and speech and language evaluations as part of standard educational programming for children with hearing loss at The River School. All data for the current study were collected as a part of these annual evaluations conducted by qualified speech–language pathologists and psychologists. Each child had one evaluation included in the current sample. Parental consent was obtained for evaluations, and verbal assent was obtained from the children for their participation. Additionally, parents were informed on enrollment that their child’s data may be used anonymously for research purposes. Children were monitored for fatigue throughout test administration, and testing was broken up over multiple sessions to facilitate motivation and energy, consistent with typical protocol in this environment. No testing was conducted beyond what was already clinically indicated for progress monitoring and treatment planning.

### 2.4. Measures

#### 2.4.1. Working Memory

Working memory was assessed by the Working Memory Index (WMI) of either the Wechsler Preschool and Primary Scales of Intelligence, Fourth Edition [25] (WPPSI-IV) or the Wechsler Intelligence Scale for Children, Fifth Edition [26] (WISC-V). Test selection was dependent on the age of the child, with children aged between 2 years, 6 months and 7 years, 3 months receiving the WPPSI-IV, and children ages 6 years to 17 years receiving the WISC-V. For children aged 6 years to 7 years and 3 months, the appropriate measure was selected based on their performance on a separately administered achievement measure [27] (Kaufman Assessment Battery for Children, Second Edition). Children who scored below average overall received the WPPSI-IV, and children who scored in the average range or above received the WISC-V.

Each Working Memory Index consists of two sub-tests that assess different functions of working memory. The WPPSI-IV WMI consists of Zoo Locations and Picture Memory sub-tests, measuring visual–spatial working memory and rote visual memory and immediate recall, respectively. The WISC-V WMI consists of the Digit Span sub-test, evaluating rote auditory working memory, and the Picture Span sub-test, assessing retention and recognition of visual information. Standard scores for the overall index were calculated in relation to the normative sample of age-matched, typically hearing children. The mean for standard scores in both measures is set to 100, with scores between 90 and 110 comprising the average range.

For the WPPSI-IV, internal consistency has been found to be between the good (0.86) and excellent (>0.90) range at the composite level [28], and test–retest reliability was also determined to be in the good range (0.84–0.89). For the WISC-V, internal consistency at the composite level was between the good and excellent range (0.88–0.93). Test–retest reliability at the index level was variable (0.75–0.94) [29].

#### 2.4.2. Language

Receptive and expressive language were each evaluated in isolation, and as a part of larger language measures. Receptive language was assessed using the Peabody Picture Vocabulary Test, Fifth Edition (PPVT-5) [30], a standardized measure that evaluates recognition of everyday words. This is carried out by presenting an array of pictures from which the child selects the image that best represents a word the examiner presents verbally. This task was administered by speech–language pathologists that are experienced in administration of this measure to children with hearing loss. This measure can be administered starting at age 2 years, 6 months, and can be given between age 90 years and beyond. Internal consistency (0.89–0.97 for Form A) and test–retest reliability (0.87–0.93 for Form A) were excellent in the normative sample. Additionally, depending on the age of the child, their language level, and clinician preference, the Receptive One-Word Picture Vocabulary Tests were administered (ROWPVT-4) [31]. Similar to the PPVT, the ROWPVT gives a child an auditory stimulus, and the child must match that one word to one of the pictures provided. The ROWPVT is given starting at age 2 years and can be administered until age 80 years. For the ROWPVT-4, internal consistency (0.94–0.98) and test–retest reliability (0.91) are excellent in the normative sample.

Expressive language was evaluated using the companion measures to the PPVT and the ROWPVT, the Expressive Vocabulary Test, Second Edition (EVT-2) [32] and the Expressive One-Word Picture Vocabulary Test, Fourth Edition (EOWPVT-4) [33]. Serving as the inverse to the receptive language testing, children are presented with images of everyday objects and must name them. The EVT-2 has excellent internal consistency (0.88–0.97 for Form A) and test–retest reliability (0.94–0.97 for Form A) [32]. The EOWPVT-4 was also found to have excellent internal consistency (0.93–0.97) and test–retest reliability (0.97) [34].

Broader language measures that provide composite scores include the Clinical Evaluation of Language Fundamentals, Fifth Edition (CELF-5) [35] and the Comprehensive Assessment of Spoken Language, Second Edition (CASL-2) [36]. These measures yield global composites that evaluate children’s’ language skills in multiple domains. The CELF-5 evaluates receptive language, expressive language, language structure and language content, all of which combine to provide a Core Language Score. The age range for this measure is from age 5 to age 21 years. The CASL-2 includes lexical/semantic, syntactic, supra-linguistic, pragmatic, expressive, and receptive language indices that contribute to the overall General Language Ability Index (GLAI). This measure can be administered to children between the ages of 3 and 21. For the current study, Core Language Score and GLAI were used as representations of overall language ability, as was the pragmatic language index of the CASL-2. This sub-scale was included as a hypothesized corollary of functional communication; if a child has the ability to use language practically, they are likely to be able to use that language functionally in their everyday lives.

The CELF-5 has been found to be a reliable and valid measure of language ability [37]. Internal consistency for each sub-test ranged from acceptable to excellent in the age bands included in this sample (0.77–0.99), and test–retest reliability ranged from acceptable to excellent (0.68–0.92). The CASL-2 had similarly strong subtest-level internal consistency (0.85–0.99), though test–retest reliability was more variable (0.65–0.90) [38].

#### 2.4.3. Socio-Emotional Functioning

Socio-emotional functioning was evaluated using the Behavior Assessment Scale for Children, Third Edition (BASC-3) [39]. This self-report measure is designed to elicit information about internalizing, externalizing, and behavioral symptoms, as well as adaptive functions. Each overall index is composed of multiple scales that target specific patterns of symptomatology or adaptive skills (e.g., inattention, anxiety, leadership). Parent and teacher forms were administered. The preschool (ages 2–5) or child (ages 6–11) form was chosen based on the age of the child. Scores on the BASC-3 are reported as T-Scores, with a mean of 50, and clinical elevations indicated at 65 and higher.

For the BASC-3, internal consistency has been found to be variable across age groups and forms of the measure (i.e., parent/teacher, preschool/child). However, all reported coefficient alphas fell in the good range or better for clinical and adaptive scales, which was the primary focus of the current analysis (Parent = 0.83–0.87, Teacher = 0.87–0.89) [39]. Test–retest reliability was found to be similar in range (Parent = 0.85–0.87, Teacher = 0.85–0.88). Construct validity was determined by the authors through factor analysis, use of the measures with children with prior diagnostic profiles, and through comparison to existing and validated measures evaluating similar constructs (e.g., autism spectrum disorder and attention-deficit hyperactivity disorder rating scales). It is important to note that this measure, among others in this study, was not normed on children with hearing loss.

### 2.5. Analytic Plan

Data were deidentified prior to analysis. Pearson correlations examined associations of working memory and language to parent- and teacher-reported domains of socio-emotional functioning. Multiple linear regression was used to examine relationships between variables significantly correlated at the bivariate level. Analyses were conducted using RStudio in R version 3.6.3 [40]. Due to the small sample size, power for sub-group analyses was limited. As a result, this study could not separate groups based on demographic variables, such as gender and racial/ethnic background.

## 3. Results

Descriptive statistics were calculated for all study variables and are presented in Table 2 and Table 3. Of note, the mean standard score for each language or working memory measure fell within the “average” range (SS = 90–110). However, statistical analysis was based on individual scores rather than the mean, which is reported here to characterize the sample. Standard deviations were larger than would be expected (SD = 12.17–22.95, expected = 10). Ranges were consistent with the normative sample, as were median scores. Means and standard deviations for BASC-3 variables were comparable to the normative sample, with all sample means and medians falling within a standard deviation of the expected mean of T = 50.

### 3.1. Bivariate Associations

#### 3.1.1. Working Memory

Working memory scores were positively associated with teacher-reported functional communication (r = 0.37). Working memory was negatively associated with teacher-reported attention (r = −0.31), atypicality (r = −0.49), hyperactivity (r = −0.44) and withdrawal (r = −0.42; see Table 4). Working memory scores were positively associated with parent-reported adaptability (r = 0.37) and functional communication (r = 0.33). Working memory was negatively associated with parent-reported attention (r = −0.46), atypicality (r = −0.35), and hyperactivity (r = −0.31; see Table 5). Working memory scores were not significantly correlated with any of the language variables in the current study.

#### 3.1.2. Language

Teacher-reported inattention was significantly and negatively related to expressive vocabulary (r = −0.34; see Table 5 for correlations between language and socio-emotional functioning), receptive vocabulary (r = −0.44), and pragmatic language (r = −0.55). Teacher-reported anxiety was significantly and positively related to core language (r = 0.44). Teacher-reported functional communication was associated with all language measures, and teacher-reported social skills scores were positively related to receptive vocabulary (r = 0.42). Parent-reported inattention (see Table 5 for parent form correlations with language) was significantly and negatively related to receptive vocabulary (r = −0.38) and core language (r = −0.40). Parent-reported atypicality was associated with receptive vocabulary (r = −0.33). Parent reported social skills were also associated with receptive vocabulary (r = 0.33).

### 3.2. Regression Analysis

Socio-emotional domains that were associated with working memory and language were the subject of regression analysis. These analyses included other correlated factors, specifically other socio-emotional domains and demographic variables, if they were associated. Gender, age at identification, and configuration of devices were not significantly correlated with any of the variables found to be related to working memory and language. Age at testing was found to be correlated with both parent (r = 0.39, *p* < 0.05) and teacher (r = 0.38, *p* < 0.05) report of functional communication, but not with the other variables (see Table 6). This is consistent with children with hearing loss developing stronger communication skills with age.

Due to the number of correlated factors, regression model fit was determined using a backward stepwise regression model. Backward elimination models iteratively remove variables from a model until optimal model fit is achieved. Model fit in this case was determined using the Akaike Information Criterion (AIC), an estimation of predictive error. A lower AIC indicates less error, and thus improved model fit [41]. Stepwise regression was conducted using the “stepAIC” function in R [42].

When predicting working memory, the initial model included all socio-emotional variables correlated with working memory at the bivariate level (see Table 7). Optimal model fit (AIC = 142.05, R^2^ = 0.391) was achieved when including teacher ratings of hyperactivity (b = −0.38), withdrawal (b = −0.45) and parent report of adaptability (b = 0.45). The same procedure was conducted to predict Core Language. Optimal model fit (AIC = 159.46, R^2^ = 0.614) included teacher report of functional communication (b = 1.06) and anxiety (b = 1.07), and parent report of adaptability (b = 0.79). Optimal model fit (AIC = 199.67, R^2^ = 0.506) for expressive vocabulary was achieved with parent reported adaptability (b = 0.56) and teacher reported functional communication (b = 1.74). The model for pragmatic language initially included teacher reported attention problems (b = −0.65) and teacher rating of functional communication (b = 1.53). This was the optimal model (AIC = 100.23, R^2^ = 0.544). All optimal model fit is described in Table 7.

Model fitting for receptive vocabulary was complicated by the high correlations between functional communication and receptive vocabulary. Optimal model fit was achieved with only parent (b = 0.65) and teacher (b = 1.42) report of functional communication in the model (AIC = 183.96, R^2^ = 0.639).

## 4. Discussion

Results of the current study supported our hypotheses, such that higher working memory was associated with lower scores in inattention, hyperactivity, withdrawal, and atypicality, as well as higher scores in functional communication. Stronger working memory skills were hypothesized to predict, overall, a more positive report of socio-emotional outcomes, which was consistent with our findings. Stepwise regression models yielded information regarding the best model fit for predicting working memory and language. Working memory was predicted by report of socio-emotional functioning, as were language variables.

Teacher report of functional communication served as a predictor for most of the language variables, which is consistent with what would be expected given the content of the functional communication scale. Parent report of adaptability was also a predictor in several of the models, suggesting that flexibility may bolster working memory capacity and language outcomes. Teacher reported hyperactivity was a predictor of working memory, which is consistent with the theory that executive functioning deficits would impact behavioral regulation and working memory capacity. Core language was also predicted by teacher report of anxiety, which may be related to likelihood to speak; if a child is anxious and worried about using their language, they would score lower on core language measures. Teacher report of inattention was a predictor of pragmatic language scores, suggesting that children with challenges in focusing and attending to social and academic opportunities would then struggle to use their language effectively in those situations.

Functional communication was the main predictor of receptive vocabulary scores. This may be important to note with regard to use of this measure in children with hearing loss. The functional communication scale includes items related to getting needs met and advocating for oneself, neither of which is predicated on language use. In this sample, the children who had the strongest ability to communicate were the ones with the strongest receptive language, whether they communicated linguistically or through gestures, actions, and approximations. The BASC-3 functional communication index may be useful in tracking receptive language capacity, as well as broader communication skills.

It is likely that the items of the BASC-3, when used with children with hearing loss, capture how children present when working memory or auditory processing is challenging (e.g., shutting down, acting odd). Moreover, if a child is unable or hesitant to respond verbally, they are likely to appear withdrawn. As would be expected, reported functional communication was significantly and positively related to language outcomes. However, report of inattention was negatively related to language outcomes, potentially highlighting the importance of identifying and separating which components of a child’s presentation are related to behavioral challenges, auditory access, and working memory.

Many professionals work with only a few children with hearing loss in their lifetimes, and even fewer are provided the training to do so effectively. Information from the current study demonstrates the need for specific training when psychologically evaluating children with hearing loss or providing diagnoses. For example, some of the behaviors comprising the atypicality scale are common for children with hearing loss, especially when their auditory access and language skills are still developing. Many measures used to evaluate children rely on auditory-only instructions and sometimes auditory-only activities, few of which are explicitly normed on children with hearing loss. The presentations and cognitive profiles of these children are unique and may be diagnostically misleading to someone with infrequent exposure to work with children with hearing loss.

Socio-emotionally, children with hearing loss can experience a variety of behavioral challenges, just like their typically hearing peers. When conducting differential diagnosis for a child with hearing loss, one should always include their hearing loss and early developmental history, particularly through the lens of the connectome model. Etiology of hearing loss can also be important to understand. Hearing loss can be associated with other complex syndromes, like Usher syndrome, and other medical diagnoses that impact other areas of functioning. For example, choosing the appropriate strategies for a child with hearing loss may be different depending on whether they have comorbid visual impairment. Multi-sensory approaches to learning, such as programs including tactile, auditory, and visual input together, can be effective in bridging this gap.

### 4.1. Limitations

A primary limitation of the current study is the sample size. Power of these analyses is limited by sample size, though large samples in a low incidence population are rare. Though the sample is large enough for analyses to approach normality, generalizability of these results is limited by the size and demographic characteristics of the sample (e.g., geographic location, school setting). Caution should be used in generalizing these results to other children and they should primarily be considered as a first step toward more robust research. Data analysis was conducted retrospectively on existing data collected from routine evaluations, which may introduce additional variables that would be accounted for in a prospective study. Some characteristics of the sample (e.g., listening devices, severity of loss, age of identification) were varied, which might also impact generalizability. Etiology of hearing loss might also play a role in interpretation of the results of this study, though for many of the children, etiology of their hearing loss was not known.

### 4.2. Future Directions

The current study is limited by including one time point, rather than following these children through time. Future research will include data for cognitive, language, and socio-emotional functioning measures over time. Additionally, increasing the sample size in future studies will be important for statistical power and the ability to analyze this type of data by demographic subgroups. Intervention is another avenue for potential research. With working memory and language found to be related to socio-emotional functioning, interventions promoting each domain would support the others. Implementation of trials of such interventions would give valuable information regarding the relationships of the current study variables. Collection of data on interventions and measures normed on typically hearing children would serve to validate their use in children with hearing loss as well.

### 4.3. Call to Action

A call to action as a result of the current study is for more training and available materials for learning about children with hearing loss to be created and made available for all providers. Knowing whether a child is amplified, what kind of device they use, and if they use sign language to supplement their spoken language is crucial to providing successful support in the classroom and home environments. Having available materials that allow practitioners to gain greater understanding can aid families in getting what they need through the educational system without needing an advocate or legal representative. Navigating elementary school and developing an individualized education plan (IEP) is challenging, particularly when evaluators are unfamiliar with the challenges and needs of a child with hearing loss. Dissemination of this information and promotion of curiosity and growth can improve access to appropriate provisions for all children in schools.

Inclusion is also an important criterion to consider when some children have comorbid psychological disorders or medical conditions. If a child with hearing loss is also diagnosed with ADHD, it is possible that they will be separated into a more restrictive environment than the general education classroom. This deprives the child of the opportunity to access the typically developing peers, who can serve as language models and support their language development. Giving children access to accommodation, while still keeping them in a general education classroom when possible, exemplifies the least restrictive environment, and children have been consistently shown to develop stronger language skills when immersed in a general education environment. Providing children with the tools to advocate for themselves in that kind of classroom will increase their success; if they did not hear a direction or could not hold it in their working memory, the only way to get that information is to ask for it.

Overall, children with hearing loss have unique profiles with regard to working memory, language, and socio-emotional functioning. Though there are variations in how auditory deprivation impacts the developing brain, some trends and predictive capacities emerged. With regard to working memory, children with higher working memory scores are less likely to struggle with behavioral, emotional, and cognitive regulation, key tenets of executive functioning. Future directions for this research include interventions that target building working memory capacity through multi-modal instruction, potentially in sensitive or critical periods, that may bolster language development as well.

## Figures and Tables

**Table 1 jcm-13-01637-t001:** Relevant Demographic Characteristics of Sample.

		*n*	%
Age at Evaluation (Months)			
	M (SD)	62 (23.61)	-
	Mdn	57	-
Gender			
	Female	23	65.71
	Male	12	34.29
Racial/Ethnic identity			
	Asian/Asian American	8	22.86
	Black/African American	4	11.43
	Latinx	1	2.86
	White	19	54.29
	Middle Eastern/North African	3	8.57
Home Language			
	Spoken English Only	24	68.57
	English/Sign Language	4	11.43
	English/other spoken language	7	20.00
Age of HL identification (months)			
	M (SD)	13.03 (13.99)	
	Mdn	8	
Etiology of HL			
	Hereditary	15	42.86
	Congenital Infection	2	5.71
	Postnatal Infection	2	5.71
	Unknown	18	51.43
Listening Device Configuration			
		Bilateral, Unilateral	
	CI	18, 0	51.43
	HA	9, 4	37.14
	CI/HA (bimodal)	4	11.43

**Table 2 jcm-13-01637-t002:** Descriptive Statistics for Language and Working Memory.

	M	SD	Mdn	Range
Core Language/GLAI	98.94	18.83	103	(48, 121)
Receptive Vocabulary	95.74	21.45	95	(55, 132)
Expressive Vocabulary	93.57	22.95	96	(55, 132)
Pragmatic Language	103.11	18.17	108	(66, 140)
Working Memory Index	97.6	12.17	98.5	(74, 116)

**Table 3 jcm-13-01637-t003:** Descriptive Statistics for BASC-3 Variables.

	Parent (T-Score)			Teacher (T-Score)		
	M	SD	Mdn	M	SD	Mdn
Activities of Daily Living	46.97	10.01	47			
Adaptability	52.57	7.96	53	51.83	8.01	51
Aggression	48.20	6.05	45	48.14	7.26	46
Anxiety	49.23	7.04	48	46.57	7.78	46
Attention	48.63	8.97	48	49.71	9.40	49
Atypicality	49.37	9.38	46	48.14	6.37	46
Conduct Problems	45.00	5.58	44	46.60	7.07	44
Depression	48.11	8.18	46	47.91	8.02	45
Functional Communication	46.77	10.25	48	46.46	8.27	47
Hyperactivity	49.00	7.10	48	47.60	7.70	46
Leadership	51.40	10.15	52.5	50.09	8.57	46
Learning Problems				47.73	8.11	46
Social Skills	51.20	8.83	53	50.66	8.05	51
Somatization	48.83	10.07	47	46.71	6.35	45
Study Skills				48.64	7.79	47
Withdrawal	50.20	8.55	50	50.66	10.66	47

NB: Activities of Daily Living is not included as a scale in the Teacher Report Form, and Learning Problems and Study Skills are not included as scales in the Parent Report Form.

**Table 4 jcm-13-01637-t004:** Statistically Significant Report Form Associations with Working Memory.

	Teacher				Parent	
Scale	Atypicality	Functional Communication	Hyperactivity	Withdrawal	Adaptability	Inattention
Correlation	−0.49	0.37	−0.44	−0.42	0.37	−0.46
*p*-value	*p* < 0.01	*p* < 0.05	*p* < 0.05	*p* < 0.05	*p* < 0.05	*p* < 0.01

**Table 5 jcm-13-01637-t005:** Statistically Significant Report Form Associations with Language.

		Core/GLAI	PPVT/ROWPVT	EVT/EOWPVT	Pragmatic
Teacher	Inattention	-	−0.44	−0.34	−0.55
	Anxiety	0.44	-	-	-
	Functional Communication	0.6	0.77	0.69	0.68
	Social Skills	-	0.42	-	-
Parent	Inattention	−0.4	−0.38	-	-
	Atypicality	-	−0.33	-	-
	Adaptability	0.43	-	0.39	-
	Functional Communication	0.6	0.7	0.6	-
	Social Skills	-	0.33	-	-

NB: All associations in the table above are significant at least *p* < 0.05.

**Table 6 jcm-13-01637-t006:** Language and Working Memory—Correlations with Demographic Variables.

	WM	Core GLAI	Rec.	Exp.	Pragm	Gender	Age	Age at ID
WM								
Core/GLAI	0.37							
PPVT/ROWPVT	0.26	0.79 *						
EVT/EOWPVT	0.24	0.84 *	0.88 *					
Pragmatic	0.38	0.48 *	0.65 *	0.63 *				
Gender	−0.33	−0.21	0.25	0.16	0.12			
Age	0.01	0.05	0.3	0.37 *	0.22	−0.13		
Age at ID	−0.05	−0.41 *	−0.33	−0.38 *	−0.18	0.01	0.13	
Configuration	−0.35	0.06	0.24	0.11	−0.44	0.05	0.14	0.18

* Asterisk indicates significance at *p* < 0.05 at a minimum.

**Table 7 jcm-13-01637-t007:** Optimal Regression Model Fit and Equations.

	AIC	R2	Regression Equation
Working Memory	142.05	0.391	(TWithdrawal × −0.45) + (THyperactivity × −0.38) + (PAdapt × 0.45)
Core Language	159.46	0.614	(TFunctionalComm × 1.06) + (TAnxiety × 1.07) + (PAdapt × 0.79)
Expressive Vocabulary	199.67	0.506	(TFunctionalComm × 1.74) + (PAdapt × 0.56)
Pragmatic Language	100.23	0.544	(TFunctionalComm × 1.53) + (TInattention × −0.65)
Receptive Vocabulary	183.96	0.639	(TFunctionalComm × 1.42) + (PFunctionalComm × 0.65)

NB: “T” denotes Teacher Report, “P” denotes Parent Report.

## Data Availability

The data are not publicly available due to confidentiality and privacy concerns with a small sample size and the potentially identifiable population.

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
