# Peer review of "Working Memory and Language Relate to Report of Socio-Emotional Functioning in Children with Hearing Loss"

_jcm, 2024, doi:10.3390/jcm13061637_

Round 1

Reviewer 1 Report

Comments and Suggestions for Authors

I read the article titled' with great interest. The study explores the relationship between working memory and language skills and socioemotional functioning as reported by parents and teachers. Teh study employed a very small sample of a private school so the results cannot be generalised. The education standards on a private school are different to a state one. 

The methodology though was appropriate rigorous and robust. The analysis again was appropriate and carefully executed and  reported, Limitations of the study were acknowledged and implications for practice discussed. 

There are no amendments to suggest to the study and I think is an interesting piece of work. However, it should be treated as a case study and thus the results cannot be generalised to the population of hearing children. Therefore, the findings and conclusions should be treated with caution. 

Reviewer 2 Report

Comments and Suggestions for Authors

Introduction

p. 1 line 27 Early auditory deprivation can have a multitude of effects on brain development and 27 may impact cognitive capacities that extend beyond the auditory system. No reference was provided. Please provide at least one to support this statement.

 The first 3 paragraphs of the introduction provide separate information about children with hearing loss that do not flow smoothly together in a cohesive narrative. I would recommend a clearer topic sentence for paragraphs 2 and 3 so that the information is better linked.  Paragraph 4 moves to language development which again is not well linked to the 3 paragraphs before.  The authors need to consider making sure each paragraph is linked with the one before and the one after. This paragraph provides an extremely brief overview of language skills in children using cochlear implants. Expansion of this information would strengthen the paper. There is a great deal of research available which would easily allow for more depth of information. No description of speech development in this paragraph yet it is referred to in the next one.

I would recommend a section on factors that may influence communication/language development. At the moment some factors are provided in 1.1 and then again in 1.2.

I would like to see research aims added to the introduction in addition to the hypotheses already provided.

Method

Please provide the research design at the start of the method.

 Was there ethical approval for this project? Please provide the details if there were.

The language measures were appropriate and well-described.  On page 5 line 163, it states that participants receive annual assessments. It was not clear to me if the data for each participant was the results from one assessment or several. As this was not clear to me it requires further clarification.

Results

These were explained clearly with appropriate measures. It appeared that the language scores for the children were all within the normal range. As a result I found it difficult to appreciate how the other measures were related to the language scores. This should be clarified?

Discussion

Insert “rating  of” between  “teacher hyperactivity” and between “parent adaptability” (p 11 line 355). Teacher hyperactivity (p 11 line 364) is used again without “rating” or “report” which reads as if the teacher is hyperactive. Please add the additional information of report or rating wherever it is missing when the teacher is mentioned e.g. “teacher withdrawal” p.11 line 366.

The early sections of the discussion appear to be a summary of the results. A rewrite of the discussion would improve the narrative being provided. 

Reviewer 3 Report

Comments and Suggestions for Authors

This study is interesting, however there are a few factors to consider.

1-      A retrospective data analysis with a single time point and small sample size is insufficient to draw a conclusion.

2-      Females are twice as numerous as males. Did you conduct any gender-based subgroup analyses?

3-      Demographic features of the sample, such as geographic location and educational environment, may impact the generalizability of the findings.

4-      Was the etiology of hearing loss considered during the analysis?

5-      Table 4 and 5 can be merged into one table. The same for table 5 and 6

6-      How can the authors validate and normalize the response of both the parents and the teachers?

7-      The reference needs to be updated.

8-      The authors should include study limitations in the abstract.
